# Artificial General Intelligence for the Detection of Neurodegenerative Disorders

**DOI:** 10.3390/s24206658

**Published:** 2024-10-16

**Authors:** Yazdan Ahmad Qadri, Khurshid Ahmad, Sung Won Kim

**Affiliations:** 1School of Computer Science and Engineering, Yeungnam University, Gyeongsan-si 38541, Republic of Korea; yazdan@yu.ac.kr; 2Department of Health Informatics, College of Applied Medical Sciences, Qassim University, Buraydah 51452, Saudi Arabia; ahmadkhursheed2008@gmail.com

**Keywords:** artificial general intelligence, large language models, Parkinson’s disease, Alzheimer’s disease, Internet of Things

## Abstract

Parkinson’s disease and Alzheimer’s disease are among the most common neurodegenerative disorders. These diseases are correlated with advancing age and are hence increasingly becoming prevalent in developed countries due to an increasingly aging demographic. Several tools are used to predict and diagnose these diseases, including pathological and genetic tests, radiological scans, and clinical examinations. Artificial intelligence is evolving to artificial general intelligence, which mimics the human learning process. Large language models can use an enormous volume of online and offline resources to gain knowledge and use it to perform different types of tasks. This work presents an understanding of two major neurodegenerative disorders, artificial general intelligence, and the efficacy of using artificial general intelligence in detecting and predicting these neurodegenerative disorders. A detailed discussion on detecting these neurodegenerative diseases using artificial general intelligence by analyzing diagnostic data is presented. An Internet of Things-based ubiquitous monitoring and treatment framework is presented. An outline for future research opportunities based on the challenges in this area is also presented.

## 1. Introduction

Organization for Economic Co-operation and Development (OECD) nations have seen their populations decline along with an increase in average life expectancy [1]. This high average life expectancy, paired with a low fertility rate, results in a rapidly aging population, which threatens the national economy and healthcare systems. Statistics reveal that South Korea is facing the fastest decline in population growth and is on the path to becoming a super-aged society by 2025. With this increase in age, the presumed percentage of the population with dementia is expected to increase, as per a report by the National Health Insurance Service of the Republic of Korea [2]. The incidence of Parkinson’s disease (PD) and Alzheimer’s disease (AD) in South Korea has notably risen, which coincides with existing studies which correlated age with these neurodegenerative disorders (NDs) [3,4]. An estimated 8.5 million people were suffering from PD in 2019 according to the Global Burden of Diseases, Injuries, and Risk Factors Study (GBD), which furthermore caused a 100% increase in the number of deaths between 2000 and 2019 [5]. An estimated 50 million people suffer from AD, while dementia is the fifth leading cause of death across the globe [6]. Females are more likely to suffer from these diseases compared with males [7]. PD is a progressive ND which significantly degrades the quality of life (QoL) due to its impact on motor and locomotive functions. AD is a severe form of dementia which affects behavior, memory, and cognition [3,8]. In OECD countries, the economic burden of these diseases is projected to rise significantly as the population ages [9]. Early and accurate diagnosis of PD and AD is critical in managing symptoms and maintaining a stable QoL. The diagnostic tools for detecting PD and AD include clinical evaluations, imaging, and biomarker analysis [10,11]. The patients undergo an extensive clinical evaluation to identify changes in locomotion and cognitive abilities. On the other hand, radiological tools include magnetic resonance imaging (MRI) and positron emission tomography (PET) scans [12,13]. Genetic testing and biomarker analysis can further consolidate the reliability of the diagnosis of these diseases.

Artificial intelligence (AI) can assist in and improve the accuracy of the diagnostic process. The United States Food and Drug Administration (FDA) has approved several AI and machine learning (ML)-based radiological image analysis tools [14], therefore, AI and ML are coming to the forefront in the early detection and diagnosis of NDs. Neural networks (NNs) can accurately detect patterns in biomedical signals and identify objects in medical images. Convolutional neural networks (CNNs) can successfully identify abnormalities in the medial images [15]. Recurrent neural networks (RNNs) can analyze time series data to identify anomalies in the normal functions of physiological functions. Deep learning (DL) algorithms have demonstrated high accuracy in anomaly detection [16]. Additionally, large language models (LLMs) can analyze massive tranches of image and clinical datasets, clinical studies, and the research literature and assist in accurately diagnosing NDs. AI algorithms can analyze medical records, including imaging, genetics, and biomarkers, to identify patterns indicative of PD and AD [17,18].

The hope of artificial general intelligence (AGI) leading to superhuman capabilities in machines is contended in [19]. AGI is the capability of an AI model to mimic human cognitive processes. AGI allows machines to “think” and “learn”, which allows them to understand concepts and apply them across several domains. Therefore, machines can transfer the knowledge they have learned in one domain to another domain. AGI can possess four characteristics. Firstly, they can perform an unlimited number and types of tasks. Secondly, they can generate new tasks within a context, and thirdly, the agents operate using a value system which underpins task generation. Finally, they can visualize the world in a global model, which they can use to interact with the physical world [20]. LLMs are exemplified by models like ChatGPT, Gemini, and Llama, which exhibit the traits of AGI; they can gather knowledge from vast online resources and transfer their learning from these large resources to perform an unlimited number of tasks [21]. These models can perform tasks across domains such as mathematics, language processing, image, video, text generation, medicine, and software coding. Their application in the field of healthcare is profound, ranging from drug discovery to medical image processing, genomics, and clinical assistance [22].

Detecting NDs using next-generation AI models has gained traction in the past few years and attracted significant research interest. The advent of transformer models has enabled AI-based systems to understand the context of data [23]. This ability to contextualize information is critical in establishing correlation between the symptoms and history and arriving at an accurate diagnosis [24]. This review is aimed at providing insight into the prediction, early detection, and diagnosis of NDs using the latest AGI models, with a focus on PD and AD. Radiological data analysis in the context of clinical evaluations and medical tests can assist in triangulating the underlying causes and hence determining a prudent treatment plan. The American College of Radiology Data Science Institute’s AI Central database contains a list of 200 FDA-approved AI products to assist in imaging-based diagnostic products from across 100 manufacturers [25]. Approval from regulators accelerates the integration of computer-aided diagnosis in healthcare systems. A survey of DL-based approaches for detecting NDs was presented in [26]. The survey covered disease detection along with severity analysis, presenting a CNN-based methodology for ND detection. The accuracy of various DL-based approaches in identifying these diseases was between 89% and 97%. The severity of the disease was estimated with a success rate greater than 90%.

This work identifies the state-of-the-art AGI methodologies which can diagnose PD and AD using massive repositories of clinical and experimental data, including the research literature. Combining Internet of Things (IoT) with AGI can alleviate and streamline the process of monitoring at-risk individuals for diagnosis of NDs. A primer on PD and AD is presented, describing the diagnostic tools used to predict and diagnose these disorders, in Section 2. The role of AGI in the diagnosis of PD and AD is identified. To the best of our knowledge, this review is the first to present a detailed discussion on the role of AGI in ND diagnosis (Section 3). An IoT-based framework based on our previous work is presented for the ubiquitous monitoring and diagnosis of NDs in Section 4. Section 5 presents a roadmap for the future to identify the avenues for mitigation of challenges in this area. Section 6 concludes the discussion.

## 2. Background

### 2.1. Neurodegenerative Disorders

NDs are a heterogeneous group of disorders characterized by the gradual deterioration of neurons, resulting in their eventual demise. Neurons are essential for operation of the brain and central nervous system, as they are the main constituents responsible for transmitting information throughout the body. The deterioration or depletion of these vital cells impacts transmission of information in the brain, leading to decreased neurological performance [27]. The global prevalence of NDs demonstrates their significant impact on public health. The most prevalent NDs, like AD and PD, affect millions of individuals worldwide, causing not just substantial degradation in QoL but also contributing to higher mortality rates. The steady course of chronic diseases typically results in extended infirmity, putting pressure on healthcare facilities, caregivers, and society [28].

PD is a common and incurable ND characterized by tremors, freezing of gait (FoG), and slurred speech patterns, among other symptoms. While the exact reason is still unknown, it is now acknowledged that the emergence of PD is affected by a complex interplay of genetic and environmental factors which effect essential cellular processes, suggesting that environmental factors do not exclusively cause the disease. The intricate etiology of this condition renders it challenging to diagnose during the initial phases and effectively manage symptoms during the advanced stages. Although the exact cause of most cases of Parkinson’s disease is unknown, and they occur randomly, specific genes have been identified as the culprits of rare familial forms of the disease when they undergo mutations. The primary genes linked to these forms are α-synuclein (SNCA), parkin, leucine-rich repeat kinase 2 (LRRK2), PTEN-induced putative kinase 1 (PINK1), and DJ-1 [29,30].

AD is the primary cause of dementia. The presence of *A*β plaques and hyperphosphorylated τ tangles characterizes Alzheimer’s disease. However, a better understanding of the disorder’s complex causes has become apparent. Familial Alzheimer’s disease, a rare early-onset form, is caused by genetic mutations in the amyloid precursor protein and presenilin genes which affect Aβ metabolism. Sporadic Alzheimer’s disease, on the other hand, is common and is caused by a complex interaction of age and disease promotion factors, as well as the fundamental processes of amyloid and τ pathology. Amyloid-β42 and τ proteins are established biomarkers in cerebrospinal fluid. Emerging biomarkers, including amyloid β oligomers and synaptic markers, provide new insights [31,32].

Scientists find NDs intriguing due to their complex pathophysiology and genetic, environmental, and biological factors which impact their progression. Understanding NDs comprehensively is crucial to discovering and implementing effective therapies and interventions. The diagnosis of NDs might prove challenging, as the symptoms sometimes overlap with those of other conditions, hence complicating the initial differentiation between various disorders. The frequent similarity in clinical presentation causes delays in accurate diagnosis of the condition and can complicate selecting suitable treatment solutions. The diversity of NDs, such as Huntington’s disease, motor neuron disease, and multiple system atrophy, presents unique challenges and possibilities for research, particularly in comprehending the fundamental mechanisms which lead to neuron degeneration [33].

MRI and fluorodeoxyglucose PET are widely recognized imaging modalities for the diagnosis of AD, as well as for detecting structural alterations and diminished cerebral function. Amyloid PET is becoming increasingly popular for visualizing amyloid plaques, while τ PET identifies τ protein tangles, offers valuable insights for distinguishing between different diagnoses, and selects patients for trials. Cerebrospinal fluid (CSF) biomarkers which find amyloid-beta and τ proteins are two other methods which can be used. Newly developed blood-based biomarkers like plasma phosphorylated tau are also used. Modern brain imaging methods like high-resolution structural MRI, functional MRI (fMRI), and diffusion tensor imaging (DTI) help doctors make more accurate diagnoses by finding small changes in the brain. Genetic testing is used to identify risk factors, such as the APOE ϵ4 allele, while AI improves data analysis and allows for personalized diagnosis and treatment planning. These technological advancements contribute to the early and precise diagnosis of Alzheimer’s disease, as well as enhancing its treatment and control. Advanced in vitro systems, such as the “organoid-on-a-chip”, are increasingly important for developing diagnostics and treatments of NDs. These systems integrate microfluidic chips and organoids to represent human disease states accurately. This unique technique surpasses prior animal models by replicating NDs’ pathogenic features more accurately and enhancing human situations’ modeling. These tools benefit academic research and pharmaceutical drug development, enhancing our comprehension of ND processes and the precision and effectiveness of therapeutic methods [34].

Despite numerous research efforts, precise diagnosis and treatment of NDs continue to be challenging. This challenge is partly due to the diseases’ delayed manifestation of clinical symptoms, by which time significant brain damage is often already apparent. Advancements in molecular biology, neuroimaging, and computational neuroscience have opened up new possibilities for identifying and treating conditions at an early stage. AI and ML are valuable tools in research for analyzing complex information and predicting the progression of diseases [35]. These capabilities are essential for providing personalized patient treatment. Furthermore, genomic medicine’s potential to make significant progress in comprehending the genetic foundation of NDs is a beacon of hope. This understanding could potentially lead to the identification of novel therapy targets. Another vital area of study is biomarker identification, which offers valuable insights into the evolution of illnesses and responses to therapy [36]. The current progress in medical health research, encompassing nanotherapeutics, stem cell therapies, ML and DL methodologies, and and enhanced imaging techniques, is commendable in the fight against NDs [37]. Utilizing data-driven methodologies can greatly boost medical progress by enhancing the precision and cost-efficiency of preventing, detecting, and treating NDs [33,38].

Although abundant research is available from over 50,000 publications, particularly with mentions of NDs in the abstracts cataloged by the Web of Science until June 2024, The rapid pace of scientific advancement necessitates ongoing updates and changes. The persistent scholarly focus is important and necessary for deepening our understanding, sharpening clinical procedures, and improving patient outcomes. Because of the serious effects of these diseases, it is critical to constantly gather new research and technological developments to make progress in the field of diagnosis, treatment, and management.

### 2.2. AI and AGI in Disease Diagnosis

AI can be perceived as the capability of a machine to imitate “intelligent” behavior [39], which is characterized by the ability to learn, reason, perceive, and understand language. The evolution of AI has accompanied the evolution of computers. AI is integrated into diverse disciplines, including computer science, mathematics, biomedical sciences, finance, operations research, economics, and statistics [40]. AI models are based on algorithms which can process data, learn from it, and make decisions. AI algorithms are designed to identify patterns in data, make predictions, and optimize learning processes [41]. AI algorithms are trained over a dataset to identify patterns based on the key data features. ML is a subclass of AI which trains a machine to imitate the human learning process through training, acting, and error correction. Based on the approach to learning, ML can be broadly divided into the following categories, illustrated in Figure 1. These approaches form the basis of training-based classification in another AI subclass called DL.

#### 2.2.1. Supervised Learning

Supervised learning involves training the ML model using labeled datasets. Labeled datasets contain data points which are contextualized. Data are annotated or “labeled”, which is performed to classify the data. Supervised learning maps the inputs to outputs based on an annotated dataset containing mapped input-output data pairs [42]. Supervised learning ignores any relationship between the data points which are used as predictors; instead, it builds up a model based on the conditional probability of an event, given the specific predictors are present. Despite the higher data annotation cost, supervised learning is highly accurate and efficient. However, an inherent bias due to biased labeling can occur. Moreover, labeled datasets are quite limited in number compared with unlabeled datasets, especially due to the involvement of the additional step of labeling the data. Labeling can be performed manually, automatically, or with hybrid systems, thereby increasing the cost of supervised learning models [43]. In biosciences, understanding causality due to these predictors is critical. Some of the applications of supervised learning algorithms are medical image classification, spam identification, and cyber security. Supervised learning algorithm examples include support vector machines, naïve Bayes, logistic regression, random forests, and decision trees [44]. Their efficacy in medical signal and image analysis is established in the research literature [45].

#### 2.2.2. Unsupervised Learning

Unsupervised learning is capable of training with unlabeled data. The model attempts to identify the patterns in a dataset and understand the relationship between them. The training datasets are not annotated, which leads the model to determine the latent structure in the datasets [46]. Clustering and association algorithms fall into this category [47], where the AI model determines the relationship between the data points and groups them according to the identified latent structure. Examples of unsupervised learning include k-means, autoencoders, and principal component analysis. Unsupervised learning applications in bioinformatics use large datasets of gene and deoxyribonucleic acid (DNA) sequences. The model maps out the relationship between them and assists in predicting DNA sequences and new drugs [48].

#### 2.2.3. Semi-Supervised Learning

This class combines the best of two worlds: supervised and unsupervised learning. This hybrid approach depends on labeled and unlabeled data for training [49]. A small sample of labeled data can be used for supervised learning, and a large dataset of unlabeled data can be used for unsupervised learning during the training period [50]. Based on the aggregate loss of supervised and unsupervised learning, the model can optimize its learning. Some of the strategies used in semi-supervised learning include entropy minimization for adjusting the border of the data clusters near the low-density areas and the proxy-label approach, which assigns proxy labels to unlabeled datasets based on supervised learning and repeats training on pseudo-labeled and labeled datasets [51,52]. Other approaches include consistency regularization, graph-based, and holistic methods. These methods have found widespread application in biomedical image analysis [53].

#### 2.2.4. Reinforcement Learning

Reinforcement learning (RL) eliminates the use of datasets for training by learning from experience. RL learns through experience. An agent running the model uses an interaction-based approach to reinforce its learning. An agent interacts with the environment by taking random actions during the training phase, collecting rewards and penalties. The accumulated rewards are used to predict future outcomes due to the set of possible actions [54]. This procedure eliminates the need to train the agent using datasets and enhances operational efficiency. Q-learning is a simple yet powerful RL technique which maintains a Q-table that evaluates agent–environment interaction and has been used in robotics, gameplay, and wireless communication. Other examples of RL include temporal difference methods, the state–action–reward–state–action (SARSA) algorithm, action-critic algorithms, and deep Q-networks (DQN) [55]. RL is a dynamic and powerful ML tool which has been successfully used in highly complex self-driving autonomous vehicles to find an optimal and safe trajectory. Similarly, RL is used in industries to operate autonomous robots and in healthcare to assist in robotic surgeries and medical image analysis [43,56].

AGI models are characterized by their ability to learn from vast online resources and specialized datasets. AGI models can use their learned weights to perform various tasks without further training. However, these models can be fine-tuned to perform specialized tasks as well. LLMs bear the most semblance to an AGI system [57].

### 2.3. LLMs

ChatGPT passed the Turing test, sparking a discussion on the implications of interacting with LLMs [58]. The ability of LLMs to understand natural language inputs opens up a wide scope of applications. ChatGPT, Gemini, and Llama are examples of LLMs which are capable of processing human inputs and generating relevant outputs. LLMs take text inputs, generate vector embeddings for each word in the sentence, and determine the relationship between each embedding, thus understanding the context of each part of the input. The key advantage of an LLM is its capability to identify the context of a word. This capacity renders LLMs extremely powerful and suitable for several applications, such as language translations, search engines, text generation, text summarization, sentiment analysis, code generation, and grammar checks. LLMs have been credited with authorship in multiple research articles [59].

Long short-term memory (LSTM) networks, which are based on RNNs, can effectively “remember the past” to understand the present [60]. However, LLMs use transformers to remember the past and determine its impact on the future. Transformers are a class of complex NNs which are trained using unsupervised learning on unlabeled datasets.

#### Transformers

LLMs are made possible by using a transformer, essentially a multi-layer NN which can process an input in a parallel manner. A transformer consists of identically stacked NNs known as encoders and decoders or either of the two. The encoder layer comprises a multi-head self-attention mechanism and a position-wise, fully connected feedforward network. It performs positional encoding and tokenizing of the words in an input sentence. The decoder then processes this output, contextualizing the words in the sentence.

The transformer encoder comprises two key layers [61]—a multi-head self-attention mechanism and a position-wise, fully connected feedforward network—as illustrated in Figure 2. Unlike RNNs, transformers operate solely on self-attention, eliminating the need for recurrence. Self-attention allows the network to determine the importance of each word in an input sequence regardless of its position, thus understanding its context. This mechanism calculates attention weights and creates a weighted sum of the input features.

Transformers employ multi-head attention to parallelize the self-attention process, enabling multiple words to be processed simultaneously. This parallelism speeds up both training and inference. Multi-head attention generates an output from each head by analyzing the relationships between the parallel inputs. A position-wise feedforward network layer linearly transforms the output. Mathematically, for *h* inputs, each input is assigned a unique, linearly projected version of queries, keys, and values, producing *h* outputs. A fully connected feedforward network layer receives the output from multi-head self-attention as an input and performs two linear transformations with rectified linear unit (ReLU) activation. The purpose of this transformation is to introduce nonlinearity. Each encoder layer applies these transformations to all input elements but uses different weights and biases. Following these transformations, a normalization block normalizes the sum of the inputs and the sum of their outputs for both sublayers.

Transformers use positional encoding for input sequences since they lack inherent positional information. This is accomplished by adding a positional encoding vector to the input embeddings before they enter the encoding layer. The decoder uses a layer structure similar to the encoder but includes an additional masked multi-head attention layer. This layer facilitates the encoder-decoder attention mechanism, which ensures that the output sequence is created using only the previously generated tokens.

## 3. Diagnosis of Neurodegenerative Diseases Using Artificial General Intelligence

The FDA’s approval of the use of AI in the clinical domain has provided an impetus to the development of AI-based solutions to assist clinicians and researchers in diagnosis and medical research. AGI models such as ChatGPT, CoPilot, Gemini, and Llama can assist in predicting and diagnosing NDs. In general, LLMs have found their use in the detection of PD, AD, and cognitive disorders. The role of LLMs in ND diagnosis was explored in [62]. The authors presented a highly generalized discussion on the role of LLMs in detecting NDs using the AGI capabilities of LLMs and discussed future research directions. The authors identified three key areas where LLM models can contribute to the diagnosis of NDs: identifying cognitive assessment, sentiment analysis, and electronic health record (EHR) analysis. Though our analysis concurs with their classification of application areas, we present a detailed discussion of the underlying principles and algorithms available in the relevant literature. This work, compared with [62], focuses on AD and PD. The contribution of ChatGPT to detecting PD is outlined in [63]. ChatGPT offers an interactive interface which can assist clinicians and patients in analyzing patient symptoms and answering related questions.

### 3.1. EHR Analysis and Data Curation

LLMs can analyze EHRs to detect, predict, and diagnose PD and AD. The majority of EHRs are unstructured and not indexed uniformly. To train the AI models using EHRs, it is imperative to structure the data according to the FAIR principle. FAIR is an acronym for findability, accessibility, interoperability, and reusability. LLMs are capable of identifying the relationships between data based on their context. Therefore, LLMs like ChatGPT are capable of converting unstructured data into a FAIR-compliant dataset, as demonstrated in [64]. The authors used ChatGPT 3.5 to identify the contextual relationships in the embeddings, achieving 74% accuracy in structuring the neuropathological data and identifying their relationships with other disorders and symptoms. This study evaluates a small fraction of the NeuroBioBank dataset [65]. Therefore, there is an immense scope for scaling up and identifying complex relationships between genetic, pathological, and radiological data. Additionally, advanced LLM models can be utilized instead of ChatGPT 3.5. A similar methodology was employed in [66], where a “human-in-the-loop” approach was adopted. A large cohort of about 25,000 participants was used to classify the symptoms based on manually created data labels. Human curators developed data labels representing eight categories for cognitive functions, memory, language proficiency, and visual and spatial abilities. ML and NLP algorithms were deployed to classify the symptoms into eight categories. The study revealed that the most common symptoms, in order of their prevalence after the first year of PD diagnosis, were memory loss, language impairment, and an attention deficit.

### 3.2. Multi-Modal Data Interactions

An attention mechanism is the key enabler of understanding the context of the input information. A multi-modal attention-based system for the detection of AD was proposed in [67]. This work is different from [68] in that it used attention-based learning. The Multimodal Alzheimer’s Disease Diagnosis (MADDi) framework was presented to diagnose AD and differentiate it from mild cognitive impairment (MCI) in three data categories: imaging, genetic, and clinical. There is an overlap between some of the symptoms of AD and MCI, which leads to misdiagnosis. The MADDi framework can differentiate AD, MCI, and control symptoms with an accuracy of 96.88%. This study uses the Alzheimer’s Disease Neuroimaging Initiative (ADNI) dataset [69] and exploits the cross-modality interactions. Cross-modality interactions help in understanding the effects of AD on the anatomy and pathology of the nervous system. The datasets used in this study did not contain the entire “picture”. The data used for training were quite limited due to missing information when combining all three modalities. Therefore, much larger datasets are required, thereby reducing biases.

A wearable-based PD detection system was proposed in [70] which leverages the advantages of the IoT framework. The IoT framework can detect PD symptoms like tremors and FoG. It incorporates an LLM-based chatbot to interact with the patient. The patient can engage with the chatbot during and after a symptom episode and provide vocal feedback. This approach helps in developing an accurate background of the symptoms. The feedback can guide the patient to make lifestyle improvements and ensure the administering of medication. The framework is a two-stage system, with the chatbot powered by an LLM. However, an LLM may be fine-tuned to process the multi-modal data collected by the wearable device, and it can identify the interactions between the various input data types.

### 3.3. Speech Analysis

Transfer learning principles were employed in [71], where the authors used a bidirectional encoder representations from transformers (BERT)-based model, distillBERT, which is pretrained over large datasets. The learned parameters are “transferred” to extract the semantic features, which are passed on to a regressive classifier. The advantage of using the distillBERT model lies in its ability to consider the entire transcript using the attention mechanism, which converts each part of the input sentence into a vector representation, and the relationship between these vectors is identified. This approach yields an accuracy of 88%. A BERT-based model was tested on the Pitts corpus, which is among the largest datasets on AD, to detect AD with an accuracy of 88.08% [72]. The LLM model analyzes speech transcripts from patient interviews describing images from the Cookie Theft datasets [73,74]. The descriptive text was analyzed to add labels to the symptoms, which were then classified by a neural network. A voting mechanism was used to group symptoms into AD and control categories. These approaches use an LLM to understand sentence structures and identify speech patterns. The speech patterns are indicative of cognition, memory, and mental coherence. A collective analysis of these parameters can assist in the detection of AD.

Perplexity in large language processing is a measure of certainty of the predictions made by the model using the context of the sentence preceding the predicted word. Perplexity in speech is also symptomatic of NDs, especially PD and AD. The authors of [75] took a novel approach to compare the perplexities in speech transcripts obtained from diagnosed AD patients in the DementiaBank dataset [73]. By comparing the perplexities in AD patients’ speech and curated speech transcripts using a neural network-based language model, the authors could demonstrate a reliable diagnosis of AD. An alternative methodology was presented in [76], which extracts the acoustic features of the speech recordings from the Alzheimer’s Dementia Recognition through Spontaneous Speech only (ADReSSo) dataset. The speech waveforms were tokenized and then analyzed using the BERT model. Pretraining embeddings on an LLM model yielded higher detection performance than text embeddings. However, in [77], the authors converted the speech recordings to text and then predicted the probability of mild cognitive impairment developing into AD within 6 years. This study used speech transcripts from the Framingham Heart Study (FHS), which is among the most comprehensive study cohorts. The results revealed that speech features can reliably determine disease progression without regard for demographic features.

Speech analysis constitutes a significant proportion of the available research literature on detecting AD using an LLM. The Alzheimer’s Dementia Recognition through Spontaneous Speech (ADReSS) dataset challenge held during the INTERSPEECH 2020 conference invited researchers to propose diagnostic algorithms from speech or language data and a method to infer the Mini-Mental Status Examination (MMSE) score [78,79]. This challenge inspired most LLM-based methodologies to predict AD using the ADReSS dataset. Transfer learning implies that the model can use the learned outcomes of one dataset on other datasets or tasks. The speech is transcribed and analyzed using AGI models to identify traces of degradation of speech capabilities, loss of words, and irregular speech patterns.

Several LLM-based approaches were proposed during this challenge to identify key features and distinguish AD from the control. Interestingly, the authors of [80] found the frequency of the filler word *uh* to be higher than *um* in AD patients, identifying them as hesitation markers. The BERT and Enhanced Representation through Knowledge Integration (ERNIE) models were able to interpret pauses during speech. The ERNIE model demonstrated the best performance at 89.6% accuracy on the test set of the ADReSS dataset. The authors of [81] showed that a fine-tuned BERT model outperformed the ML algorithms. The addition of curated datasets can further enhance the performance of AD detection. The BERT model, enhanced by novel embeddings and vector representation, can be used to reliably detect and predict the occurrence of AD [82]. The authors of [82] exploited the mel-frequency cepstral coefficients (MFCCs), which were curated to identify the presence of AD. The authors of [83] combined the analysis of the textual and acoustic features to identify cognitive impairments in the absence of any one type of input. In [84], the authors compared several combinations of non-attention- and attention-based algorithms to fulfill the challenge goals. They implied that a DL-based methodology for curating the inputs can enhance the overall detection performance.

### 3.4. Key Outcomes

The state-of-the-art approaches can be roughly classified into the above-stated three categories, which are based on the type of input data and the available datasets for training. The performance of LLM models can be conditioned based on the availability of the datasets, which can fine-tune the models to obtain domain-specific knowledge. Curating these datasets plays a critical role in the detection performance as well as the efficiency of the training process. The accuracy of AGI in predicting and detecting NDs greatly depends on disease modeling and the training datasets. Gender and ethnic features should be considered during disease modeling and model training to further enhance the accuracy, which might be affected by model generalization for a disease type [85]. Lack of control data or ground truths, incomplete data, and a lack of ethnic or gender diversity can compromise the performance of the AGI model, as NDs may present uniquely in different patients, owing to differences in anatomical and genetic features stemming from race and sex [86,87]. Multi-modal data can enhance the model’s understanding of the dependencies between the different biomedical data, such as clinical, radiological, genetic, and pathological. This broad understanding can be extrapolated to predict and diagnose other disorders in addition to the ND considered in this study. The key outcomes of this study are summarized in Figure 3.

## 4. Internet of Things Framework for the Diagnosis and Treatment of NDs

The convergence of the IoT and healthcare can relieve current healthcare systems of excessive pressure as the average age of a population increases [88]. A synergy between next-generation wireless networks and AI can alleviate the unnecessary burden of monitoring patients at healthcare facilities and help in continuous monitoring of the patients to generate alerts in case of emergencies [89]. As stated in Section 2.1, PD leads to resting tremors which increase in severity with the disease’s progression. To detect and suppress tremors, we propose an online framework for detecting and suppressing resting tremors in patients with PD [90]. Deep brain stimulation (DBS) introduces a controlled electrical current to cancel the electrical impulses causing tremors [91]. However, sustained stimulation can negatively impact brain function. Therefore, the authors of [90] used AI to determine appropriate doses of electrical stimulation.

This work proposes an AGI-based detection and personalized medicine framework for NDs. The authors of [70] proposed an IoT-based LLM chatbot to accurately determine symptoms. However, this work further proposes a detection and treatment framework leveraging AGI to generate alerts and initiate countermeasures. Our proposed framework is illustrated in Figure 4. Wearable sensors connected to a central controller over a low-latency wireless local area network (WLAN) can continuously collect health data and deliver feedback in real time [92].

The proposed framework can identify the time series electromyogram and positional sensor signals, such as those from accelerometers and gyroscopes, to establish the relative positions of limbs and detect resting tremors. Training the LLM using multi-modal datasets enables the model to establish a correlation between various input signals. An alert is generated by a central controller such as a smartphone, which communicates with the sensors and DBS delivery module using a quality of service (QoS)-compliant Wi-Fi network when a tremor is detected. An appropriate tremor suppression signal is generated and delivered using the DBS module. The model is capable of determining the suitable levels of stimulating electric current by analyzing the deviations from the baseline electrical activity. This ensures that excessive stimulation is avoided to prevent long-term harm. However, there are critical challenges to be addressed before such a system can be approved for use in patients.

## 5. Future Research Directions

AGI will have a profound impact on healthcare systems, especially in the context of early detection and prediction of diseases, including NDs. However, with the current status of state-of-the-art methodologies, several research challenges need to be addressed before mainstream adoption and acceptance in healthcare systems is achieved. The challenges AGI is currently facing can be summarized in the following categories.

### 5.1. Challenges in LLM Operation

Training an LLM is a computationally and environmentally expensive process. The cost to acquire the required hardware and energy needed for operation is still incredibly steep. Sustainability is a key challenge to address, as training an LLM requires energy-intensive hardware such as graphical processing units (GPUs) and tensor processing units (TPUs). The development of low-energy hardware is an active research area. NVIDIA’s Blackwell platform is capable of running LLMs for medical applications while reducing power consumption by about 25% [93]. The new hardware can also contribute to a reduction in training time. However, algorithmic changes to the LLM architecture and adaptive learning rate scheduling can reduce the training time. Moreover, removing redundant parameters, known as model pruning, and lower precision representations for weights and activations, known as quantization, can improve the inference times. Transfer learning can use related knowledge to perform tasks with lower inference times. Once an LLM is trained, and input is provided, it may generate unrelated, inaccurate results. LLMs are often seen as ”black boxes”, making it difficult to understand how they arrive at specific outputs. Additionally, LLMs also tend to generate completely fabricated results with no basis, referred to as hallucinations. These can be mitigated by implementing retrieval-augmented generation (RAG) [94]. RAG verifies the generated output using an external basis, thereby preventing misinformation. Occasionally, an LLM may provide highly disconnected results for similar input prompts, also known as prompt brittleness. Prompt engineering and reinforcement learning from human feedback (RLHF) can overcome this challenge [95].

### 5.2. Challenges in Datasets

Training an LLM for generating meaningful output requires a large corpus of data, and in preparing an LLM for a specialized task, fine-tuning is performed using highly specific datasets. However, the available datasets for healthcare applications are mostly unstructured and disjointed. Therefore, these datasets must be standardized to a common format. The majority of health records are available as handwritten notes and are not annotated for training. Curating these datasets using automation and defining a common standard can significantly improve the training data quality and the performance of information retrieval. Moreover, the training datasets are highly skewed in favor of specific genetic and economic demographics due to economic and social inequalities. This bias can significantly impact the accuracy of results, as NDs can present differently in different genetic and gender groups [96]. Additionally, there is a requirement for coherence between multi-modal datasets to accurately classify the symptoms and eliminate the probabilities of false positives and false negatives. LLMs can be used to understand the relationships between the symptoms, curate the datasets, and create annotations for the symptoms. These synthesized datasets can be used for fine-tuning and RAG.

### 5.3. Ethical and Privacy Challenges

Healthcare data are strictly protected under law to maintain the privacy of the patients. The risks associated with the misuse of private medical data are significant and can lead to identity theft, insurance fraud, discrimination, and fatal harm. Therefore, regulations such as the Health Insurance Portability and Accountability Act (HIPAA) and the General Data Protection Regulation (GDPR) are in effect in the United States and Europe, respectively [97]. These protections mandate that explicit consent is provided by the individuals to publicly access their data, and they protect their identities by implementing stringent privacy-preserving measures. Therefore, it is challenging to balance redaction and useful information to train LLMs effectively. Significant effort and computational resources are required to implement high-level encryption and privacy-preserving measures. Differential privacy can mitigate the risks of reverse inference by adding statistical noise within the datasets to protect individual identities while effectively conveying useful information. Additionally, a framework must be set up to constantly monitor the access privileges and eliminate any loopholes in data sharing.

## 6. Conclusions

The aging population trend is expected to face an increased risk of NDs such as PD, AD, and dementia. However, the symptoms of these health issues may overlap and result in misdiagnosis. Increased adoption of AGI in healthcare applications has inspired confidence in using AGI for the prediction and early diagnosis of NDs. LLMs hold the best semblance to AGI. Therefore, this work reviews the state-of-the-art LLMs used for the early detection and diagnosis of PD and AD. A primer on PD, AD, and AI is presented, followed by a discussion on the use of LLMs on different classes of datasets to diagnose PD and AD. We propose an IoT-based framework to improve the QoL of patients by monitoring and managing symptoms. An outline of the challenges and future research directions is also presented.

## Figures and Tables

**Figure 1 sensors-24-06658-f001:**
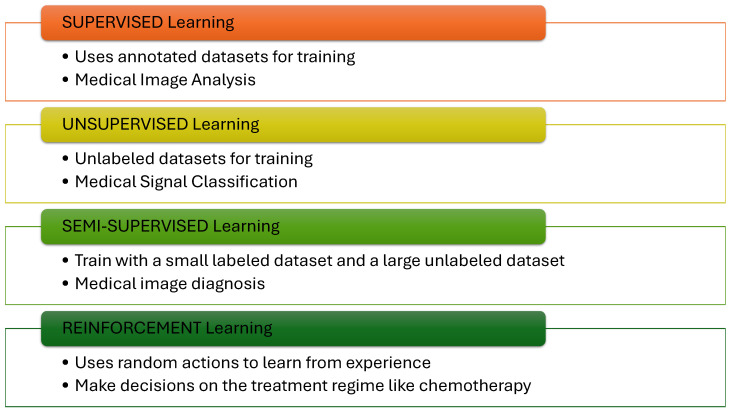
Types of AI algorithms.

**Figure 2 sensors-24-06658-f002:**
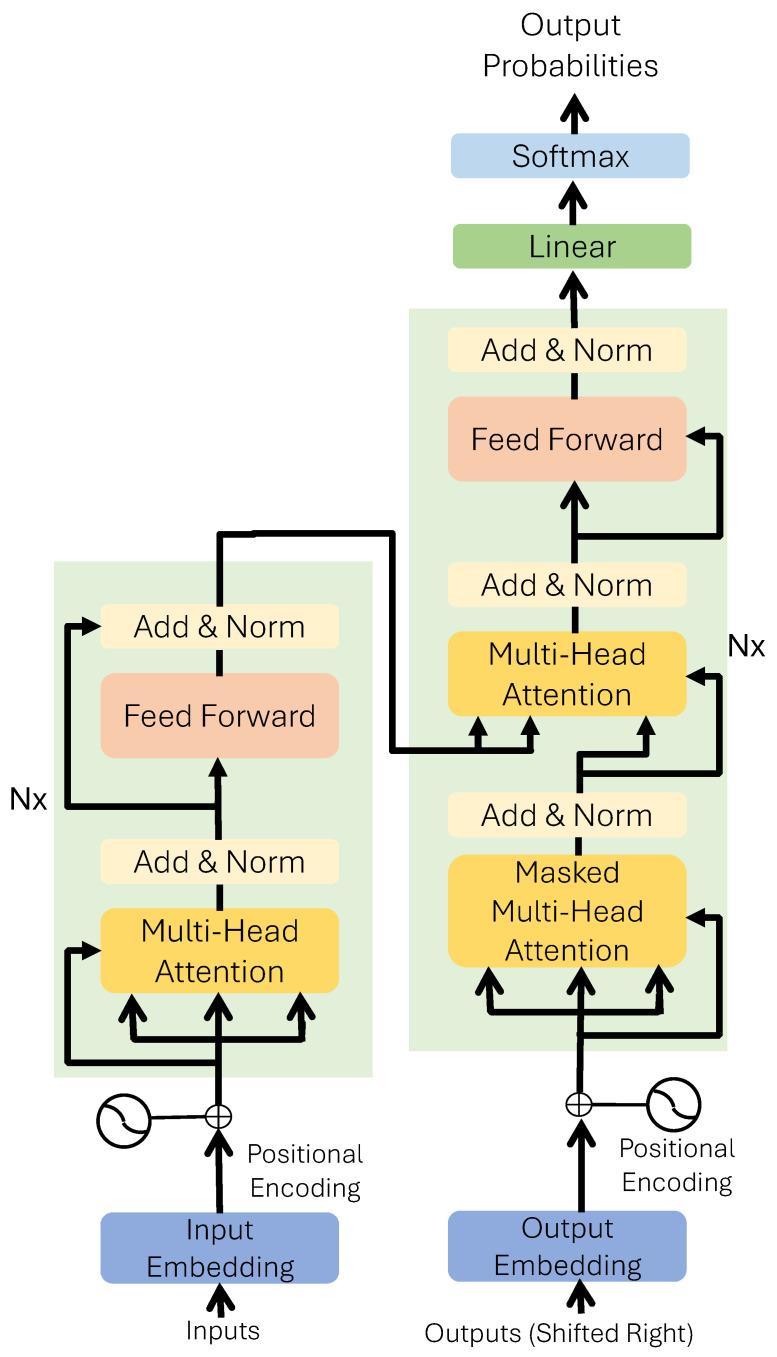
Transformer architecture described in [61]. An encoder and a decoder with positional encoding enable the attention mechanism.

**Figure 3 sensors-24-06658-f003:**
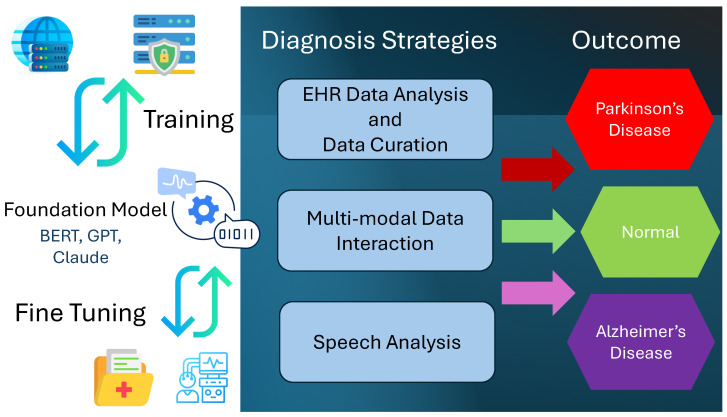
Diagnosing NDs using AGI. Summary of Section 3.

**Figure 4 sensors-24-06658-f004:**
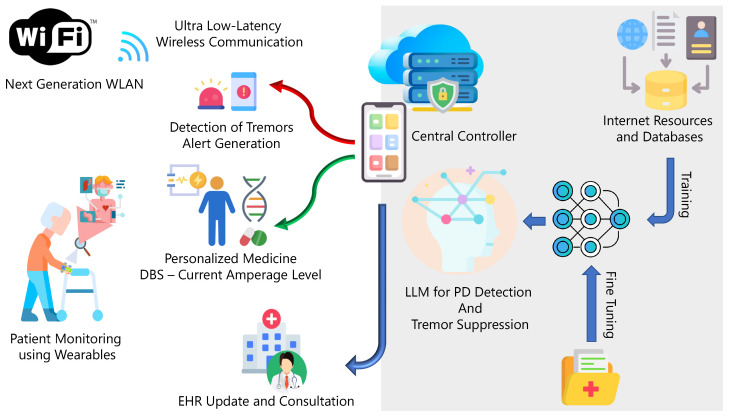
Monitoring and managing symptoms by converging the IoT and AI. An IoT-based framework is proposed to detect PD symptoms and alleviate symptoms using controlled administration of DBS and drugs.

## Data Availability

Not applicable.

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
