# Peer review of "Artificial General Intelligence for the Detection of Neurodegenerative Disorders"

_sensors, 2024, doi:10.3390/s24206658_

Round 1
Reviewer 1 Report
Comments and Suggestions for Authors
I read the review by Qadri et al. with interest. The use of AGI certainly has a great future in the early diagnosis of neurodegenerative diseases. The review is well structured and easy to read, and will undoubtedly be of interest to a wide range of researchers. The detailed list of abbreviations, which unfortunately cannot be avoided, but which may not be familiar to readers, is very useful.
Minor comments:
1) Caption to Fig. 2. It should be noted that this figure is borrowed from the article [55].
2) Page 1, lines 19-20. "aging demography", it would be better to rephrase it somehow. After all, demography is a science.
Comments on the Quality of English LanguageEnglish language is fine
Author Response
- Summary
We are extremely thankful for reviewing our manuscript and providing valuable insights to improve the quality of the manuscript. We have tried our best to address the minor comments in the reviewer report to the best of our ability.
- A point-by-point response to Comments and Suggestions for Authors
I read the review by Qadri et al. with interest. The use of AGI certainly has a great future in the early diagnosis of neurodegenerative diseases. The review is well-structured and easy to read, and will undoubtedly be of interest to a wide range of researchers. The detailed list of abbreviations, which unfortunately cannot be avoided, but which may not be familiar to readers, is very useful.
Minor comments:
Comment 1: Caption to Fig. 2. It should be noted that this figure is borrowed from the article [55].
Response 1: We acknowledge that the image presented in Fig. 2 has been borrowed from an earlier publication as this technology has been open-sourced by the original author’s organization (Google). We had acknowledged the source in the caption as pointed out in the comment. However, we have used the same architecture to redraw the image ourselves and retained the acknowledgment by citing the image source.
We have updated the figure and the caption.
“Transformer Architecture described in [55]. An encoder and a decoder with positional encoding enable the attention mechanism.”
Comment 2: Page 1, lines 19-20. "aging demography", it would be better to rephrase it somehow. After all, demography is a science.
Response 2: We appreciate this comment to add more clarity to the intended sentence. We have replaced the word demography with population as this was our intention in the first place as
“The high average life expectancy paired with a low fertility rate results in a rapidly aging population which threatens the national economy and the healthcare systems”
Reviewer 2 Report
Comments and Suggestions for Authors
An interesting work devoted to a detailed analysis of modern data in the field of using artificial general intelligence to identify and predict a number of neurodegenerative disorders such as Parkinson's and Alzheimer's diseases. There are the following questions and suggestions about this work:
1. in substantiating the relevance of the work, data on the prevalence/ number of patients with Parkinson's and Alzheimer's diseases in the world should be provided;
2. Lines 36-37 should be given literary references confirming the phrase "Genetic testing and bio-marker analysis can further consolidate the reliability of the diagnosis of these diseases.";
3. in substantiating the relevance of this study, data should be provided confirming the advantages of using artificial general intelligence to identify and predict analyzed neurodegenerative disorders (for example, reducing diagnostic costs; reducing diagnosis time; improving diagnostic accuracy; more accurate forecasting of risks, etc.);
4. Lines 93-94 should provide specific data on the prevalence of PD and AD;
5. Are there any ethnic features in the use of artificial general intelligence to identify and predict the analyzed neurodegenerative disorders (features of environmental risk factors for their development, genetic characteristics, etc.);
6. Are there any sex-specific features in the use of artificial general intelligence to identify and predict analyzed neurodegenerative disorders (differences in the level of sex hormones, genetic characteristics, etc.);
Author Response
- Summary
We are grateful to the reviewer for their keen observation and constructive criticism which has helped us to enhance the clarity of our manuscript. We have tried our best to address all the comments and fulfill the reviewer’s expectations of this manuscript.
- Point-by-point Response to Comments and Suggestions for Authors
An interesting work devoted to a detailed analysis of modern data in the field of using artificial general intelligence to identify and predict a number of neurodegenerative disorders such as Parkinson's and Alzheimer's diseases. There are the following questions and suggestions about this work:
Comment 1: In substantiating the relevance of the work, data on the prevalence/ number of patients with Parkinson's and Alzheimer's diseases in the world should be provided;
Response 1: Thank you for this comment. We understand the relevance of this study can be further strengthened by providing statistics that can substantiate the claims of NDs being an area of concern. Therefore, we have used statistics from reliable sources to bolster our claims in Section 1.
“An estimated 8.5 million people were suffering from PD in 2019 according to the Global Burden of Diseases, Injuries, and Risk Factors Study (GBD)—furthermore causing a 100% increase in the number of deaths between 2000 and 2019 [5]. An estimated 50 million people suffer from AD, while dementia is the fifth leading cause of death across the globe [6]. Females are more likely to suffer from these diseases compared to males [7].”
Comment 2: Lines 36-37 should be given literary references confirming the phrase "Genetic testing and bio-marker analysis can further consolidate the reliability of the diagnosis of these diseases."
Response 2: We are grateful for the comment. PD and AD require a series of clinical, pathological, and radiological information. In this direction, we have provided relevant citations to refer to the source of this information. Zhang (2022) published in npj Parkinson’s Disease summarizes various evaluations undertaken to diagnose PD. Similarly, Arfaa et. al (2022) covers the diagnostic tools for AD.
“The diagnostic tools to detect PD and AD include clinical evaluations, imaging, and biomarker analysis [ 10,11].”
Comment 3: In substantiating the relevance of this study, data should be provided confirming the advantages of using artificial general intelligence to identify and predict analyzed neurodegenerative disorders (for example, reducing diagnostic costs; reducing diagnosis time; improving diagnostic accuracy; more accurate forecasting of risks, etc.);
Response 3: We thank the reviewer for the comment. The use of AI in disease detection, apart from NDs has been well-established in literature. Several AI models have been developed and tested for disease detection and they have achieved a high accuracy in their objective. We support our argument in support of leveraging AGI for ND detection based on the work presented by Erdaş et.al (2021). The authors have tabulated the performance of DL-based approaches in the detection of NDs. We have included the motivation for the use of AGI in ND detection in Section 1, Introduction.
“The American College of Radiology Data Science Institute's AI Central database contains a list of 200 FDA-approved AI products to assist in imaging-based diagnostic products from across a hundred manufacturers [25]. Approval from the regulators accelerates the integration of computer-aided diagnosis in healthcare systems. A survey of DL-based approaches for detecting NDs is presented in [26]. The survey covers disease detection along with severity analysis along with presenting a CNN-based methodology for ND detection. The accuracy of various DL-based approaches to identify these diseases was between 89% to 97%. The severity of the disease was estimated with a success rate of greater than 90%”
Comment 4: Lines 93-94 should provide specific data on the prevalence of PD and AD;
Response 4: We apologize for not understanding the context of the comment, especially at the location pointed out by the reviewer (lines 93-94). However, we find this comment to be similar to Comment 1 which suggests adding statistics on the prevalence of AD and PD. We have added the relevant information in Section 1, Introduction.
“An estimated 8.5 million people were suffering from PD in 2019 according to the Global Burden of Diseases, Injuries, and Risk Factors Study (GBD)—furthermore causing a 100% increase in the number of deaths between 2000 and 2019 [5]. An estimated 50 million people suffer from AD, while dementia is the fifth leading cause of death across the globe [6]. Females are more likely to suffer from these diseases compared to males [7].”
Comment 5: Are there any ethnic features in the use of artificial general intelligence to identify and predict the analyzed neurodegenerative disorders (features of environmental risk factors for their development, genetic characteristics, etc.);
Response 5: We thank the author for pointing out a very significant aspect of AGI performance for disease detection. The manifestation of NDs might be different in different ethnic and gender groups due to inherent genetic differences as reported in
- https://www.ncbi.nlm.nih.gov/pmc/articles/PMC2858583/
- https://academic.oup.com/aje/article-abstract/142/8/820/51334
However, due to disparities in the collection of digitized data from economically developing areas, the training of the AGI algorithms may be skewed, leading to inaccurate generalizations and compromising the accuracy of the AGI model. Therefore, curating the datasets is quite critical in addition to accurate disease modeling considering differences in the genetic makeup of diverse ethnic groups. We have included a clarification in section 3.4 on the effects of race on AGI model accuracy.
“The accuracy of AGI in predicting and detecting NDs greatly depends on disease modeling and the training datasets. Gender and ethnic features should be considered during disease modeling and model training to further enhance the accuracy which might be affected due to model generalization for a disease type [86]. Lack of control data or ground truths, incomplete data without lacking ethnic and gender diversity can compromise the performance of the AGI model as NDs may present uniquely in different patients owing to differences in anatomical and genetic features stemming from their race and sex [87, 88]”
Comment 6: Are there any sex-specific features in the use of artificial general intelligence to identify and predict analyzed neurodegenerative disorders (differences in the level of sex hormones, genetic characteristics, etc.);
Response 6: Thank you very much for the comment. Based on the published studies available at (Same as Comment 5),
- https://www.ncbi.nlm.nih.gov/pmc/articles/PMC2858583/
- https://academic.oup.com/aje/article-abstract/142/8/820/51334
It is found that the prevalence of PD and AD is lower in women compared to men. Paper 2 listed above revealed that women tend to be at a lower risk of developing PD and AD based on the clinical data available from the Medicare Program in the US.
“The accuracy of AGI in predicting and detecting NDs greatly depends on disease modeling and the training datasets. Gender and ethnic features should be considered during disease modeling and model training to further enhance the accuracy which might be affected due to model generalization for a disease type [86]. Lack of control data or ground truths, incomplete data without lacking ethnic and gender diversity can compromise the performance of the AGI model as NDs may present uniquely in different patients owing to differences in anatomical and genetic features stemming from their race and sex [87, 88]”
Round 2
Reviewer 2 Report
Comments and Suggestions for Authors
The authors made all possible adjustments to the article and provided all comments on the questions/suggestions expressed in the review. The article is recommended for publication.